# Indian Ocean warming as a driver of the North Atlantic warming hole

Shineng Hu [1,2✉] & Alexey V. Fedorov[3,4]

Over the past century, the subpolar North Atlantic experienced slight cooling or suppressed warming, relative to the background positive temperature trends, often dubbed the North Atlantic warming hole (NAWH). The causes of the NAWH remain under debate. Here we conduct coupled ocean-atmosphere simulations to demonstrate that enhanced Indian Ocean warming, another salient feature of global warming, could increase local rainfall and through teleconnections strengthen surface westerly winds south of Greenland, cooling the subpolar North Atlantic. In decades to follow however, this cooling effect would gradually vanish as the Indian Ocean warming acts to strengthen the Atlantic meridional overturning circulation (AMOC). We argue that the historical NAWH can potentially be explained by such atmospheric mechanisms reliant on surface wind changes, while oceanic mechanisms related to AMOC changes become more important on longer timescales. Thus, explaining the North Atlantic temperature trends and particularly the NAWH requires considering both atmospheric and oceanic mechanisms.

[1] Lamont-Doherty Earth Observatory of Columbia University, Palisades, NY, USA. [2] Division of Earth and Ocean Sciences, Nicholas School of the Environment, Duke University, Durham, NC, USA. [3] Department of Earth and Planetary Sciences, Yale University, New Haven, CT, USA. [4] LOCEAN/IPSL, Sorbonne University, Paris, France. ✉email: shineng@ldeo.columbia.edu

The North Atlantic warming hole (NAWH) is a robust, observed feature of the North Atlantic temperature trends in the past century[1]. Various observational datasets generally agree on the warming pattern over the North Atlantic with surface waters south of Greenland warming less than the surrounding oceans, although actual temperature trends depend on dataset (Fig. 1). A similar NAWH pattern also emerges in global climate simulations using historical forcing or future $CO_2$ emission scenarios[2,3].

The causes of the NAWH however remain debatable. In principle, sea surface temperature (SST) changes could result from both oceanic and atmospheric processes. A number of previous studies argue that the slowdown of AMOC and the resultant weaker ocean heat transport convergence into the North Atlantic, could potentially explain the NAWH seen in climate simulations[2–9]. However, to what extent this mechanism has already been operating in reality remains unclear, in part due to the absence of sufficiently long instrumental measurements of AMOC strength. A recent modeling study identified a similar NAWH pattern in atmospheric simulations coupled to a mixed-layer ocean wherein the effects of large-scale ocean circulation were suppressed, suggesting that the NAWH could be driven solely by atmospheric processes[10]. In another example, in-situ measurements reveal that wind-induced cooling was probably responsible for the substantial heat loss in the Irminger Sea during the winter of 2014–2015[11]. Similarly, the enhanced storminess associated with a northward shift of the North Atlantic jet stream is shown to effectively cool the ocean mixed layer locally, potentially contributing to the NAWH[12].

In the current work, we describe a novel mechanism for the formation of the NAWH that is related to the remote effects of Indian Ocean warming. Following our observational analysis and coupled model simulations, we argue that the historical NAWH can potentially result from the strengthening of overlying surface westerly winds and the latter can be remotely driven by the concurrent Indian Ocean warming. Our model-based estimates suggest that this mechanism can explain most of the historical NAWH. However, we also find that on longer timescales the Indian Ocean warming will warm the subpolar North Atlantic by strengthening the AMOC, and henceforth, we treat this analysis as a case study of the relative importance of wind versus AMOC changes for the NAWH.

## Results

**Observational evidence.** To quantify the strength of this anomalous cold pattern in the North Atlantic, we define an NAWH index as the SST difference between the small and big blue boxes in Fig. 1a (Methods). The NAWH index began to decrease in the 1940s, reducing by over 1 °C in half a century, recovered slightly after 1990, and then continued to decrease in the 2000s (Fig. 2c).

While the NAWH was developing, surface westerly winds over the subpolar North Atlantic significantly strengthened since the

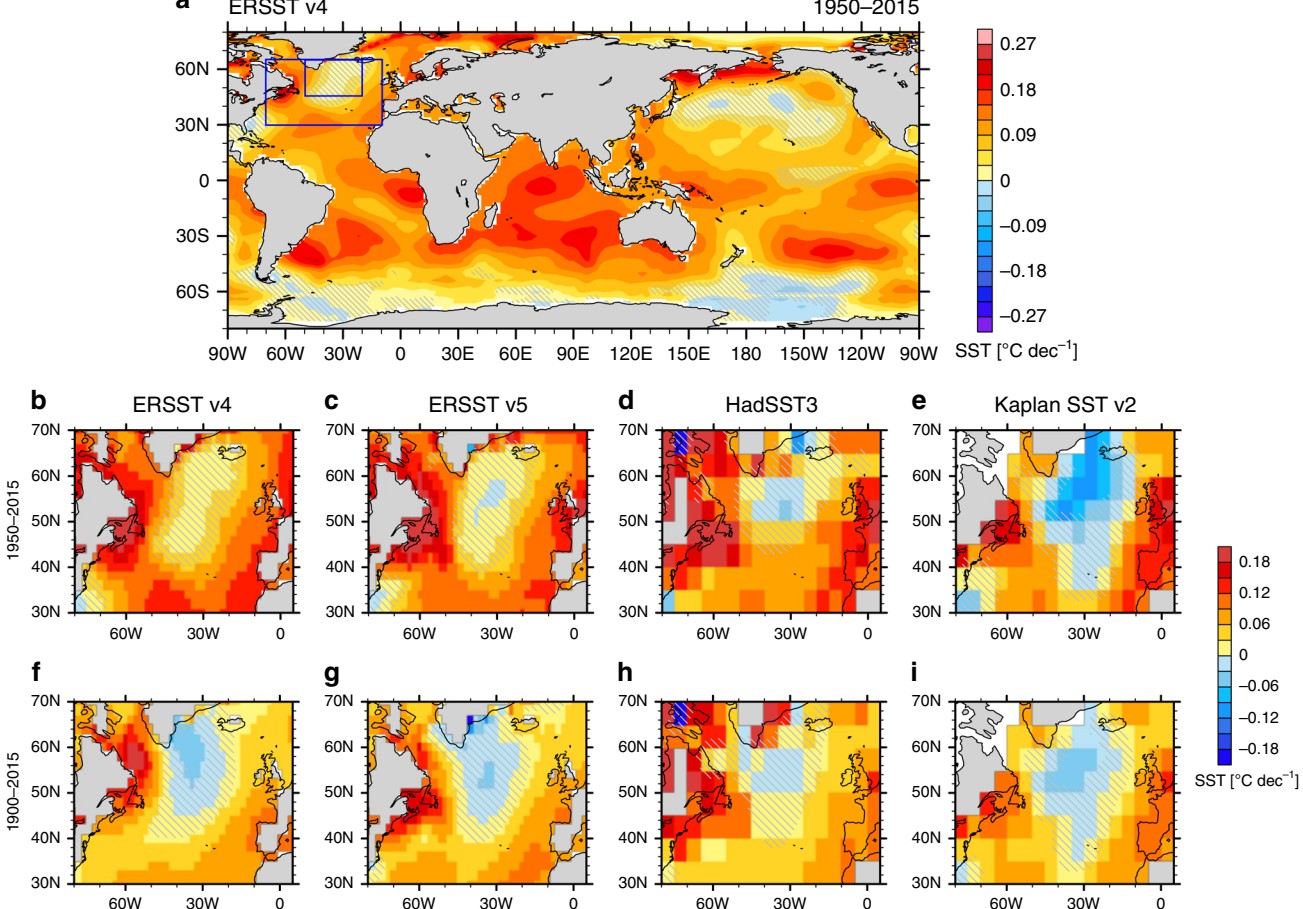

**Fig. 1 Historical warming trends. a** Global pattern of observed trends in annual-mean sea surface temperature (SST; units: °C/decade). The North Atlantic warming hole (NAWH) index is defined as the difference between average SST for the two blue boxes. **b–f** As in panel (**a**) but focusing on the extratropical North Atlantic and using different datasets and time periods. Gray hatches highlight the areas of lower statistical significance (*p*-values of Student's *t* test above 0.05).

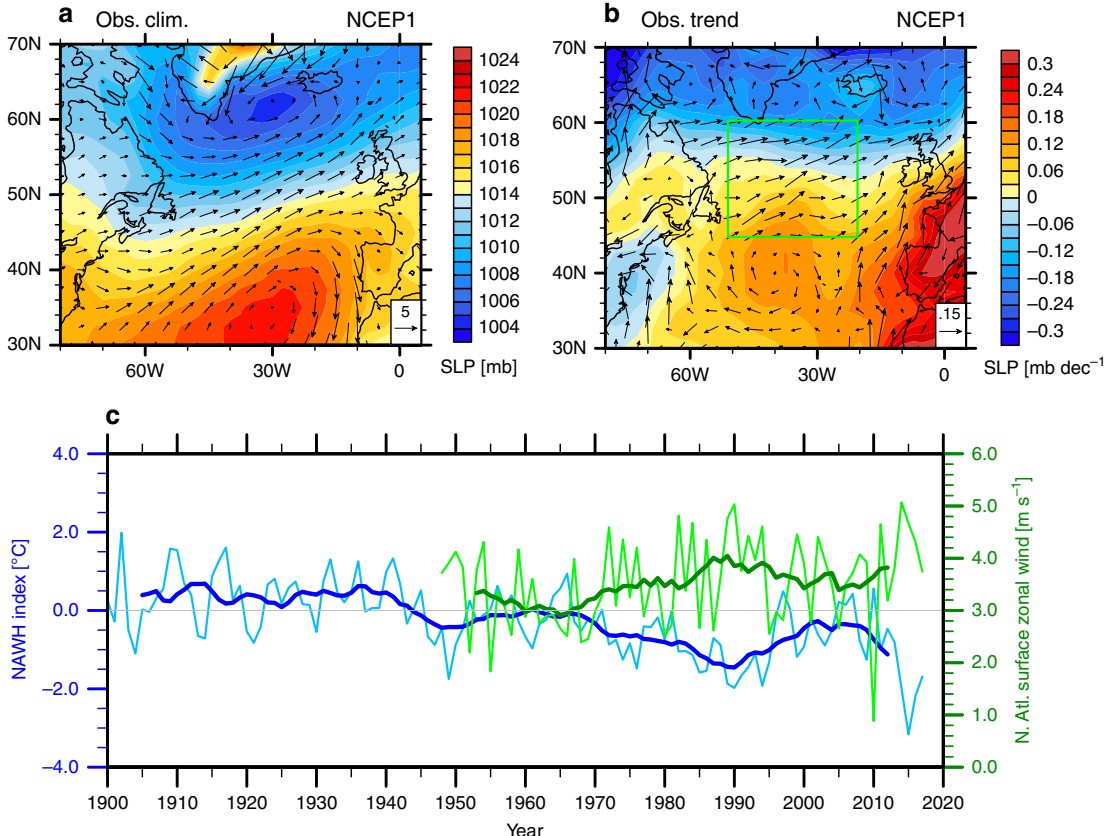

**Fig. 2 The observed North Atlantic wind strengthening and the North Atlantic warming hole (NAWH) index. a** 30-year climatological mean and **b** long-term trends during 1950–2015 in sea level pressure (SLP; units: mb) and surface wind vectors (units: m s$^{-1}$). **c** The annual-mean NAWH index (blue) and surface zonal winds (green) averaged within the green box in panel (**b**); the heavy lines represent 11-year running means. The surface wind and sea level pressure data are from NCEP/NCAR Reanalysis 1; the NAWH index is calculated based on ERSST v4.

1940s. Averaged over the subpolar North Atlantic (45°N–60°N, 50°W-20°W), surface zonal wind has strengthened by about 0.7 m s$^{-1}$ (Fig. 2c) and zonal wind stress has increased by about 0.02 N m$^{-2}$ (not shown) since 1950. This wind strengthening is associated with the strengthening and a slight northward shift of the meridional dipole of sea level pressure (SLP; Fig. 2a, b). Such changes are part of the well-recognized upward trend in the North Atlantic Oscillation (NAO) index observed during the second half of the 20th century[13–15].

Stronger surface westerlies over the subpolar North Atlantic could potentially cool the underlying ocean by driving anomalous Ekman flow that transports cold high-latitude water southward. We can obtain a rough estimate of the anomalous Ekman transport-induced heat flux using the equation $Q'_{Ek} = (c_p \tau'_x/f)(\partial T/\partial y)$, where $c_p$ is the specific heat of seawater, $\tau'_x$ is the surface zonal wind stress change, $f$ is the Coriolis parameter, and $\partial T/\partial y$ is the climatological meridional SST gradient. A long-term change of $\tau'_x$ by 0.02 N m$^{-2}$ would give rise to a negative $Q'_{Ek}$ of about $-3$ W m$^{-2}$ (assuming $c_p = 3850$ J kg$^{-1}$ °C$^{-1}$, $f = 1.1 \times 10^{-4}$ s$^{-1}$, and $\partial T/\partial y = -4 \times 10^{-3}$ °C km$^{-1}$) or equivalently a cooling effect of about $-0.2$ °C year$^{-1}$ (assuming an ocean mixed layer depth of 100 m). At the same time, the higher surface wind speed can perturb the surface heat budget by enhancing surface turbulent heat loss from the ocean to the atmosphere. Those dynamical and thermodynamical effects could operate together and lead to the cooling of the ocean mixed layer underneath the strengthened surface westerlies[16].

We next analyze the observed surface heat fluxes (positive if downward) over the same period. Changes in net surface heat flux are dominated by surface turbulent heat flux (mainly latent heat flux), while the contribution of surface radiative flux is relatively small (Fig. 3; Supplementary Fig. 1). The surface turbulent heat flux shows a slightly positive trend over the warming hole, implying that the ocean has been gaining heat over this region since the mid-20th century (Fig. 3c). This surface heat gain is presumably balanced by the oceanic cooling effect due primarily to the anomalous southward Ekman transport. The reduction in turbulent heat flux is mainly caused by the decrease of SST, i.e. the thermal damping effect (Fig. 3a), which opposes and overwhelms the effect of surface wind strengthening. The dominance of the thermal damping effect as a negative feedback for the temperature change is even clearer to the southwest of the warming hole where the signal is large. Therefore, the analysis on the surface heat flux change alone that mixes the forcings and feedbacks might provide limited or even misleading information on the origin of surface temperature changes.

Nevertheless, observational evidence for the possible atmospheric forcing of the NAWH does exist. We find that the surface zonal wind averaged over the subpolar North Atlantic tightly correlates with the NAWH index since the late 1940s; stronger surface winds tend to be associated with a stronger (i.e. more negative) NAWH (Fig. 2c). The negative correlation is particularly evident on decadal and longer timescales ($r = -0.90$; see the thick lines in Fig. 2c) and may also exist on interannual timescales during some years. This finding implies that potentially the observed trend of NAWH and its decadal-to-multidecadal variability could be forced by the overlying surface winds. In a coupled system, one could certainly argue that it might be the

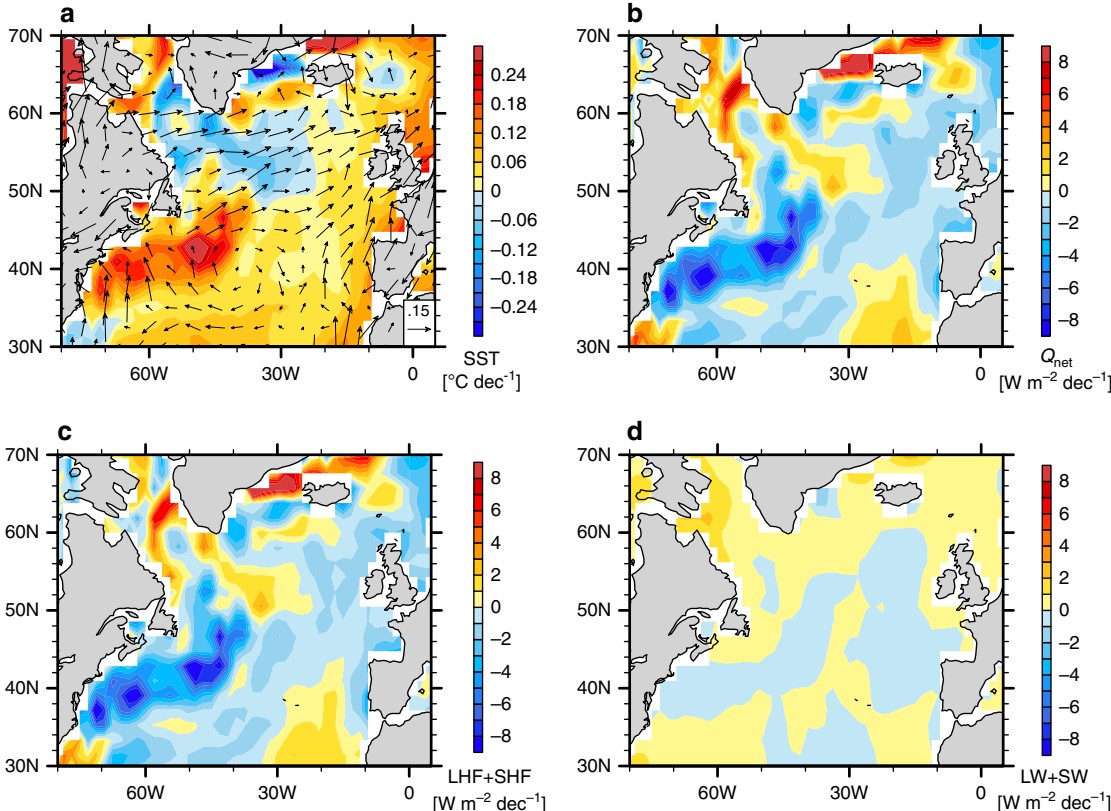

**Fig. 3 Observed climate trends over the North Atlantic.** The plot shows long-term trends during 1950–2015 in (**a**) SST (°C decade$^{-1}$) and surface winds (m s$^{-1}$ decade$^{-1}$), (**b**) net surface heat flux and (**c**, **d**) its components including (**c**) surface turbulent heat fluxes (latent + sensible; LHF + SHF) and **d** surface radiative fluxes (longwave and shortwave; LW + SW). All surface heat flux terms (in W m$^{-2}$ decade$^{-1}$) are defined as downward positive. NCEP/NCAR Reanalysis 1 is used.

NAWH that forces the strengthening of surface winds. However, later on we will use our climate simulations to demonstrate that local SST anomalies are not so efficient in forcing the surface westerly wind changes in the subpolar North Atlantic.

Then what could have caused the strengthening of surface westerlies? We now turn attention to the tropical Indian Ocean (TIO) whose temperature has increased by about 1 °C since the mid-1950s[17–27], at a faster rate than that of the other two tropical oceans (Fig. 1a). Previous modeling studies suggested that remote effects of enhanced precipitation and latent heat release due to the Indian Ocean warming were responsible for the concurrent, positive NAO phase shift[13–15]. Since a positive NAO is typically accompanied by the strengthening of surface westerly winds south of Greenland[28], these results have motivated us to explore whether the observed NAWH could be caused by the TIO warming through atmospheric teleconnections. At the same time, it has been suggested that on longer timescales the TIO warming can strengthen the AMOC primarily by modulating tropical hydrological cycle and increasing Atlantic salinity[26]. The strengthened AMOC implies a stronger northward ocean heat transport that can warm the North Atlantic. Below we investigate the remote impacts of Indian Ocean warming on the subpolar North Atlantic by considering both effects—the NAO wind-induced ocean cooling and the AMOC heat transport-induced warming.

**Experimental design.** To investigate these effects, we performed a suite of perturbation experiments with a coupled general circulation model (GCM), the Community Earth System Model (CESM). We firstly conducted an idealized simulation by imposing a nearly uniform 1 °C warming over the TIO while keeping the rest of the global ocean fully coupled with the atmosphere (named TIO + 1C; Methods). The warming is imposed by restoring the tropical Indian Ocean SST (30°S–30°N, 40°E–100°E with a 5° buffering zone inside the edges; see the black box in Fig. 4) to a seasonally varying climatology that is 1 °C warmer than the control run ("PI") with a relaxation timescale of 10 days. Other sensitivity simulations will be described later, and the details of experimental set-up can be found in "Methods".

**TIO-NAWH fast link via wind changes**. Here we regard the average of Years 1–40 as reflecting the fast ocean response due to the induced atmospheric changes; the 40-year duration is chosen as a compromise to be sufficiently long for statistically significant results but short enough for the generated NAWH not to decrease yet in response to AMOC strengthening. The main conclusions presented below do not change if a shorter period (e.g. 20 or 30 years) is chosen for averaging.

The warming of the Indian Ocean enhances local precipitation (Fig. 4a), and the resultant increase in latent heating acts as a tropical heat source fueling global atmospheric teleconnections (Fig. 4b; Supplementary Fig. 2). The resulting fast response to such a heating includes the strengthening and a slight northward shift of the SLP meridional dipole over the subpolar North Atlantic, accompanied by anomalous surface westerly winds south of Greenland and anomalous easterly winds over the subtropics (Fig. 5a, b). The general response patterns in SLP and surface winds are very similar to the observed trends (Fig. 2a, b), and also consistent with previous studies that identified a positive NAO-like response to the Indian Ocean warming in atmosphere-only

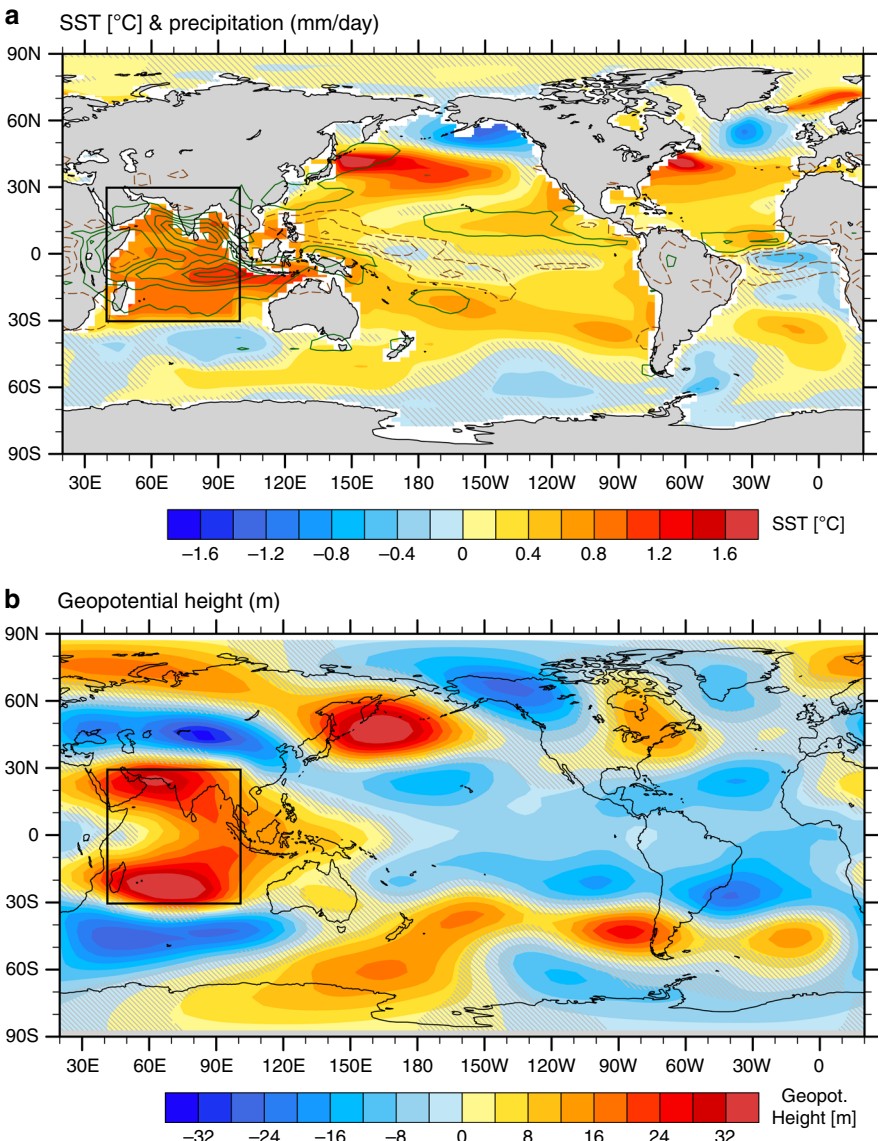

**Fig. 4 Global climate response to tropical Indian Ocean (TIO) warming.** Anomalies in (**a**) sea surface temperature (SST; units: °C) and precipitation (contours; 0.6 mm per day interval); light green and brown contours represent positive and negative precipitation anomalies starting from 0.3 mm per day and −0.3 mm per day, respectively, and (**b**) geopotential height (units: m; zonal mean removed) at level 193 of CESM's hybrid vertical coordinate, close to 200 hPa. Anomalies are averaged within the first 40 years of the integrations (i.e. the initial response) for the TIO + 1C experiment relative to the PI simulation. Gray hatches highlight the areas of lower statistical significance (p-values of Student's t test above 0.05). The box over the Indian Ocean indicates the region where SST anomalies are imposed in the perturbation experiments.

GCMs[13–15]. The broad pattern of the induced SST anomalies is also not unlike the one seen in the observations for the positive phase of the NAO[16,29].

As part of the response, the North Atlantic SST exhibits a pronounced local cooling centered at ~55°N, which contrasts the surrounding warming (Figs. 6a, 7c) and resembles the observed NAWH (Figs. 1, 3a). In principle, the subpolar cooling can be explained by two effects induced by intensified surface westerly winds (Fig. 6a). First, the wind intensification cools the ocean surface by enhancing turbulent heat fluxes to the atmosphere. However, as we argued earlier for the observations, the impact of surface wind intensification is not evident in actual turbulent heat flux changes, which are positive (i.e. downward into ocean), because these changes are largely controlled by the thermal damping of the surface cooling (Supplementary Fig. 3; cf. Fig. 6a). Second, surface westerly wind anomalies drive an anomalous southward Ekman transport of cold water, resulting in a balance between anomalous ocean heat divergence and the reduction of surface heat fluxes.

The general warming in the subtropical Atlantic is consistent with the easterly winds-induced northward Ekman transport of warm water. The enhanced western Atlantic warming around ~40° N (Fig. 6a) is probably related to the anomalous anticyclonic wind curl prevalent at those latitudes that could result in the deepening of the ocean thermocline in the west leading to warmer western Atlantic SST. As a result of all these aforementioned changes in the North Atlantic, the NAWH index experiences a rapid reduction within a few years by about 0.6 °C and stays mostly unchanged for about four decades until the AMOC starts to strengthen (Fig. 7b).

**TIO-NAWH slow link via AMOC changes**. In parallel, in response to the TIO warming, the AMOC strength gradually increases and reaches a quasi-equilibrium after about a century (Fig. 7a). In the TIO + 1C experiment, the AMOC becomes

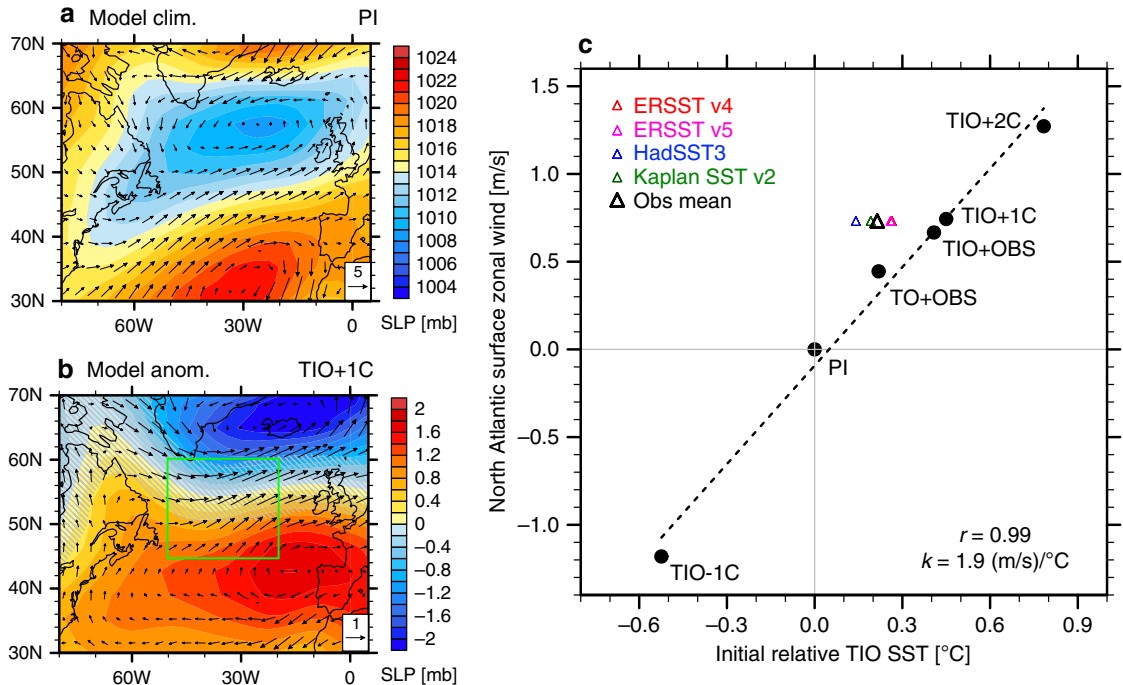

**Fig. 5 The North Atlantic surface wind response to tropical Indian Ocean (TIO) warming. a** 40-year pre-industrial (PI) climatological mean and (**b**) anomalies in the TIO + 1C experiment during Years 1–40 for sea level pressure (SLP; units: mb) and surface wind vectors (units: m s$^{-1}$). Gray hatches highlight the areas of lower statistical significance (p-values of Student's t test above 0.05). **c** Anomalies in surface zonal wind averaged within the green box shown in panel (b) in different simulations versus anomalies in the relative TIO sea surface temperature (SST). Anomalies are with respect to the pre-industrial (PI) experiment. The relative TIO SST is defined as average SST in the Indian Ocean (30°S–30°N, 40°E–100°E) minus the whole tropical ocean (30°S–30°N) value; "initial" means averaged during Years 1–40. Colored triangles indicate the observed changes during 1950–2015 (long-term trends multiplied by 66 years) for NCEP/NCAR Reanalysis winds and different SST datasets; the black triangle represents their averages. The dashed lines are linear regression curves with the correlation coefficient (r) and the regression slopes (k) also shown.

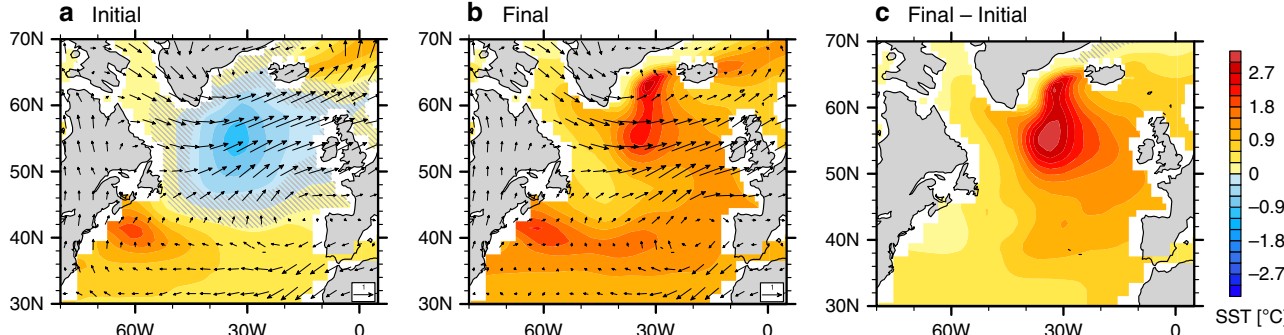

**Fig. 6 The emergence and decay of the model North Atlantic warming hole (NAWH) in response to tropical Indian Ocean (TIO) warming.** Anomalies in sea surface temperature (SST; units: °C) and surface winds (units: m s$^{-1}$) for the TIO + 1C experiment with respect to the pre-industrial (PI) experiment, over the (**a**) initial period, (**b**) final period, and (**c**) the difference. "Initial" refers to Years 1–40, while "final" refers to Years 151–200. Anomalies in panels (**a**) and (**c**) are considered parts of the system's fast and slow responses in the North Atlantic, respectively. Gray hatches highlight the areas of lower statistical significance (p-values of Student's t test above 0.05).

stronger by ~5 Sv. The underlying mechanisms of the TIO-AMOC link are explained in detail in ref. [26]. In short, the Indian Ocean warming suppresses the Atlantic rainfall by modifying the Walker circulation along the entire equatorial belt, and the resultant positive Atlantic salinity anomalies act to strengthen the AMOC after being advected to the subpolar North Atlantic by ocean mean circulation. At equilibrium, the stronger westerly winds over the subpolar region also help maintain a stronger AMOC by enhancing surface turbulent heat loss.

The AMOC plays an important role in the global interhemispheric energy transport and its net effect is to carry ocean heat northward towards the subpolar North Atlantic. The AMOC strengthened by the Indian Ocean warming can therefore deliver more heat into the subpolar North Atlantic. As a result, the whole North Atlantic warms up as the AMOC is getting stronger (Fig. 7c). The initial cooling south of Greenland gradually vanishes and is later replaced by a basinwide warming (Fig. 6a, b) with the negative NAWH index decreasing to near zero in about eight decades (Fig. 7b). The slow response, as indicated by the difference between Years 151–200 and Years 1–40, features a pronounced warming to the south of Greenland caused mainly by the AMOC strengthening (Fig. 6c).

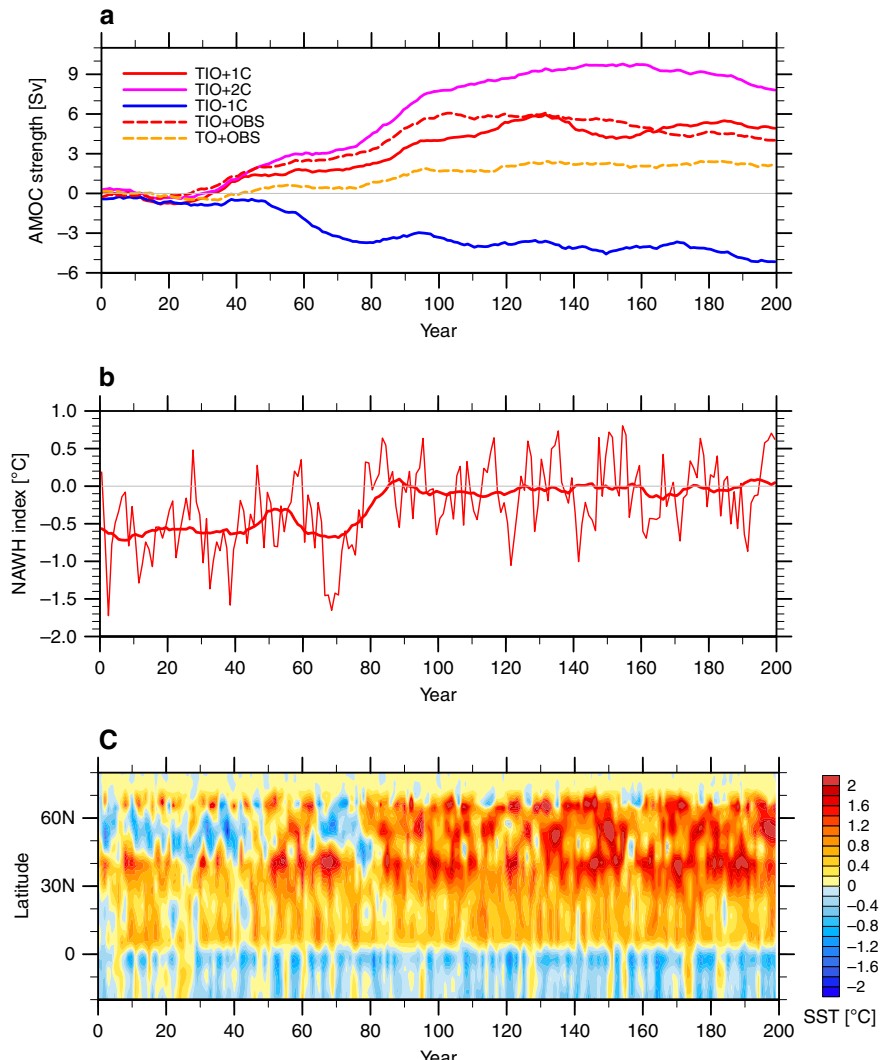

**Fig. 7 Atlantic Ocean response to tropical Indian Ocean (TIO) warming. a** Variations in annual-mean Atlantic meridional overturning circulation (AMOC) strength in different sensitivity experiments. The AMOC strength is estimated as the maximum streamfunction within 500–5500 m, 28° N to 90° N. A 21-year running mean is applied. **b** Variations in annual-mean anomalous North Atlantic warming hole (NAWH) index in TIO + 1C; the thick curve represents the 21-year running mean. The NAWH index is defined as the SST difference between the subpolar North Atlantic (45°N–65°N, 50°W–20°W) and the basinwide extratropical North Atlantic (30°N–65°N, 70°W–10°W), see Fig. 1a. **c** A Hovmoller diagram of annual-mean zonal-mean SST anomaly (°C) averaged within 60°W-0° in TIO + 1C.

It is interesting to note that the large-scale wind pattern at the final stage does not change much from the fast response (Fig. 6a, b), suggesting that the North Atlantic wind response is mostly forced by the remote tropical forcing rather than by local SST changes, and therefore the fast response in SST is driven by wind changes, rather than the other way around. For the observations, specifically for the tight correlation shown in Fig. 2c, our modeling results imply that surface wind changes are most likely the cause of the NAWH development and variations, and that the observed westerly wind intensification could indeed be driven by the concurrent TIO warming.

**Additional sensitivity experiments confirming the TIO-NAWH links.** Further, we conduct a series of sensitivity experiments to confirm the robustness of the results (Methods), and here we discuss several of them. In TIO-1C, the Indian Ocean is cooled by 1 °C—a case opposite to TIO + 1C. As part of the fast response, the surface wind field shows a negative NAO-like pattern with anomalous easterly winds over the subpolar North Atlantic and

westerly winds on its southern flank (Fig. 8a). In a direct response to the wind changes, the subpolar North Atlantic warms up while subtropical North Atlantic cools. In the slow response, the weakening of the AMOC (Fig. 6a) results in a weaker northward ocean heat transport and thus a basinwide cooling of the North Atlantic (Fig. 8b). Overall, the North Atlantic SST changes in TIO-1C generally resemble those in TIO + 1C but with an opposite sign, although some regional details may differ (cf. Fig. 6a, c).

In another sensitivity experiment, TIO + OBS, we superimposed the warming pattern actually observed in the tropical Indian Ocean during 1950–2015 (Methods). The climate response in TIO + OBS (Fig. 8c, d) is generally similar to that in TIO + 1C (Fig. 6a, c), which is not surprising given the observed TIO warming magnitude since 1950 of approximately 1 °C.

Previous studies suggest that, in the tropics, it is relative changes in local SST with respect to the tropical mean SST (sometimes called relative SST; see "Methods") that matters most for the local precipitation[26,30–32]. Therefore, we added a more realistic simulation called TO + OBS (Methods), similar to

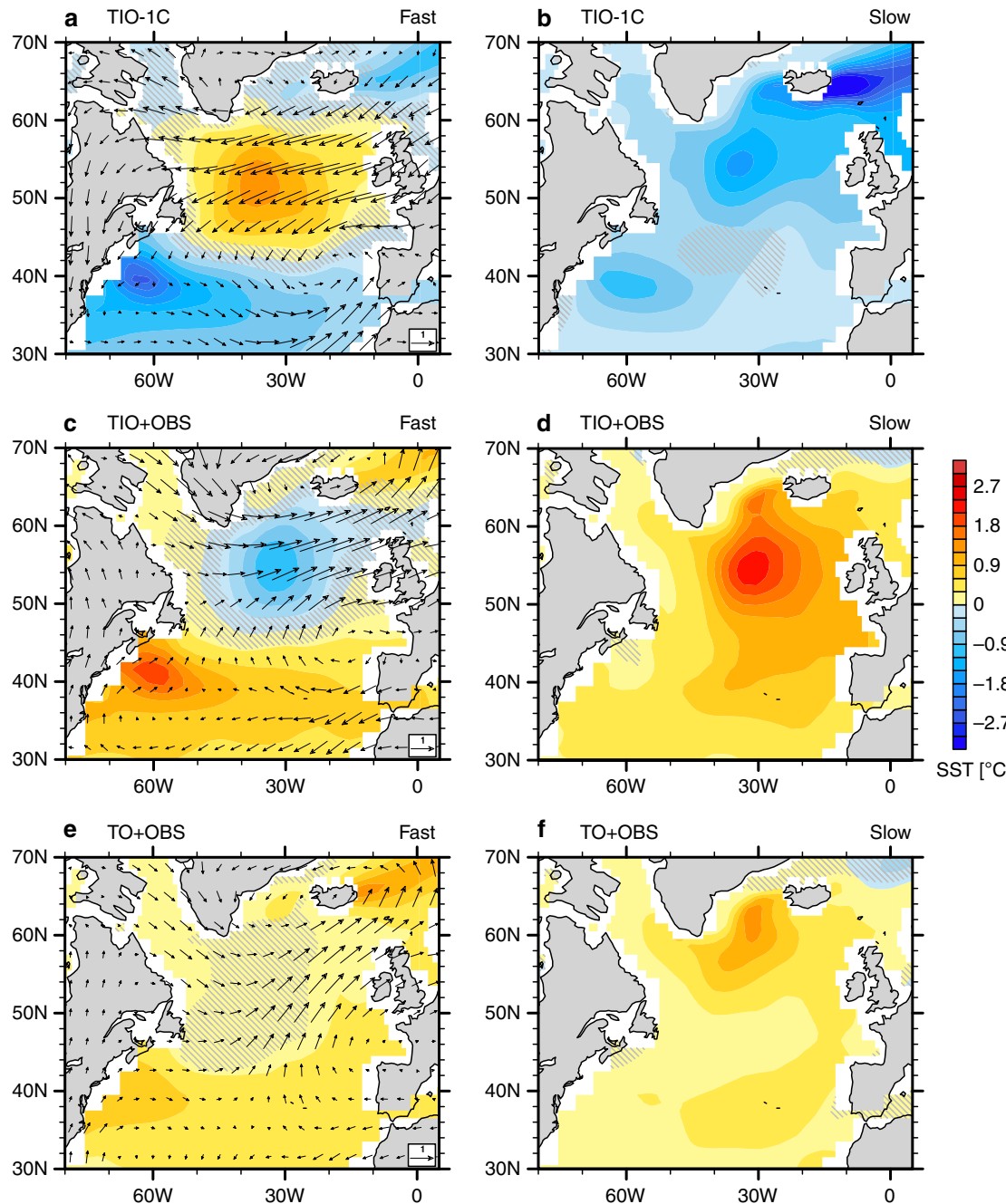

**Fig. 8 The fast and slow responses of the North Atlantic climate to tropical Indian Ocean (TIO) warming in different sensitivity experiments.**
**a**, **b** Anomalies in sea surface temperature (SST; units: °C) and surface winds (units: m s⁻¹) for the TIO-1C experiment with respect to the pre-industrial (PI) experiment; panel a is for fast response averaged over Years 1–40 (cf. Fig. 6a) and panel b is for slow response defined as the difference between Years 151–200 and Years 1–40 (cf. Fig. 6c). **c**, **d** Same as panels (**a**) and (**b**) but for the TIO + OBS experiment. **e**, **f** Same as panels (**a**) and (**b**) but for the TO + OBS experiment. See "Methods" for the details of experimental set-up. Gray hatches highlight the areas of lower statistical significance (p-values of Student's t test above 0.05).

TIO + OBS but with the observed warming imposed in all three tropical ocean basins. In this case, the relative TIO warming is weaker than in TIO + 1C or TIO + OBS due to the stronger concurrent warming of the other two tropical basins. As a result, the impacts on the AMOC (Fig. 7a) and the North Atlantic climate are also weaker (Fig. 8e, f). Nevertheless, the key features of the TIO + 1C experiment are mostly captured. For example, the anomalous subpolar westerly winds and subtropical easterly winds are both evident in the fast response (Fig. 8e). In the fast response, the North Atlantic has a suppressed warming in the

subpolar region (Fig. 8e), while in the slow response the whole North Atlantic warms with a local enhancement to the south of Greenland (Fig. 8f).

In Fig. 9 we plot the NAWH index for the fast and slow responses, separately, against the relative TIO SST for all the simulations conducted, including the TIO + 2C experiment with a 2 °C TIO warming but not described in detail here. For the fast response caused solely by atmospheric changes, the NAWH index scales tightly with the relative TIO warming (the correlation r = −0.99) as the relative warming of Indian Ocean tends to make the

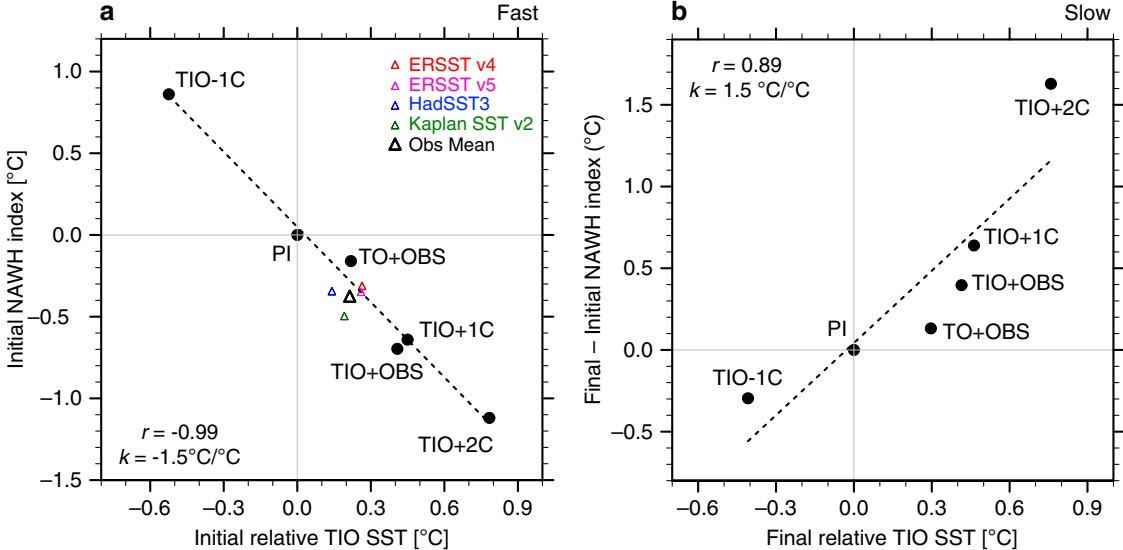

**Fig. 9 Fast versus slow North Atlantic warming hole (NAWH) responses to tropical Indian Ocean (TIO) warming.** The NAWH index versus the relative TIO sea surface temperature (SST) for the (**a**) fast and (**b**) slow responses in different simulations with respect to the pre-industrial (PI) experiment. The NAWH index is defined as average SST in the subpolar North Atlantic (45°N–65°N, 50°W–20°W) minus the broader extratropical North Atlantic (30°N–65°N, 70°W–10°W), see Fig. 1a. The relative TIO SST is defined average SST in the Indian Ocean (30°S–30°N, 40°E–100°E) minus the whole tropical ocean (30°S–30°N). "Initial" refers to an average for Years 1–40, while "final" refers to Years 151–200. Colored triangles indicate the observed changes during 1950–2015 (long-term trends multiplied by 66 years) for different datasets; the black triangle represents their averages. The dashed lines are linear regression curves with the correlation coefficients (*r*) and the regression slopes (*k*) also shown.

subpolar North Atlantic cooler than the surrounding ocean (Fig. 9a). Such a tight correlation is also found between the relative TIO SST and the North Atlantic surface zonal winds (Fig. 5c), confirming our argument that the fast response of the NAWH is induced by the intensification of surface westerlies. For the slow response associated with the remotely induced AMOC changes, the correlation is about 0.89 (Fig. 9b), as the relative TIO warming acts to warm the subpolar North Atlantic more than the ocean regions nearby, but there are some nonlinearities in the relationship, possibly related to differences between sea ice contraction and expansion and their climatic effects.

In summary, the TIO warming has opposing effects on the NAWH for the fast versus slow responses. The net impact on the NAWH is a combination of both, and its sign and magnitude depend on the relative magnitude of the fast and slow responses. Based on our model simulations, the sensitivities of the NAWH index to the relative TIO warming for the fast and slow responses are overall comparable, which makes their net long-term impact smaller due to strong compensation. It is noteworthy however that the NAWH properties (strength, location, etc.) in the fast and slow responses are constrained by different physical processes, and therefore their relative magnitudes of the two responses can vary with time, the model used, and the background climate (e.g. note the nonlinearities in Fig. 9b).

## Discussion

Has the TIO warming indeed contributed to the observed NAWH? Given the relatively long timescale of the AMOC response to TIO warming and no observational evidence for AMOC strengthening, the fast response appears to be more relevant to explaining the trends of the past half-a-century or so. Indeed, the model fast response over the North Atlantic to TIO warming (Figs. 5b, 6a) strongly resembles the observed patterns of climate trends (Figs. 1, 2b) as well as the North Atlantic wind/ SLP pattern associated with a relative TIO warming on decadal-to-longer timescales based on a linear regression (Supplementary

Fig. 4). Among all of the common features, the most pronounced ones include the meridional dipole of pressure anomalies, the strengthening of surface westerly winds north of 40°N, and the suppressed surface ocean warming south of Greenland relative to the surrounding regions.

Moreover, based on our model estimates, the fast response to the relative TIO warming during 1950–2015 can explain most of the observed NAWH anomaly (~87% on average, see Fig. 9a, but this fraction varies with the observational dataset used). In terms of surface zonal winds over the subpolar North Atlantic, about 55% of the wind intensification can be attributed to the relative TIO warming (Fig. 5c), but again this number would differ for other datasets.

Thus, the general similarities in the spatial patterns and our rough estimates above suggest that atmospheric changes related to the Indian Ocean warming can at least in part account for the observed NAWH. However, our results also indicate that on multidecadal and longer timescales atmospheric and oceanic effects of the TIO warming on the NAWH would nearly compensate each other, unless other factors act to weaken the AMOC. Therefore, AMOC slowdown should be critical for the maintenance of the NAWH on these longer timescales. In fact, in future climate projections the NAWH appears to be maintained predominantly by the robust AMOC decline[3,6,9] caused by increased ocean stratification in the high-latitude North Atlantic.

For the late 20th century and the last two decades tentative evidence suggests that the AMOC might be already weakening[4,5,33,34], but the data is either too short or based on proxy reconstructions. The emergence of the NAWH is often considered as evidence of the AMOC decline and an index based on the subpolar North Atlantic SST is used to characterize past AMOC trends. However, our results and the results of ref. [10] indicate that one has to be careful when interpreting the observed NAWH solely as a result of AMOC weakening since atmospheric effects could have played a large role in establishing the warming hole. Some studies argue that the observed NAWH cannot be forced by the atmosphere since on multidecadal timescales the North Atlantic SST leads by several

years the NAO index[34] and surface turbulent heat fluxes[35]. However, others argue that these lead-lag correlations do not necessarily shed light on causality because on such timescales SST is already in a quasi-equilibrium with the atmosphere[36]. Moreover, our results suggest that the atmospheric and oceanic mechanisms of the NAWH are not necessarily exclusive of each other, and both could operate at the same time. Thus, the relative contribution of atmospheric changes versus AMOC changes to the observed NAWH needs to be investigated and quantified in future studies.

## Methods

**Observational datasets.** We use the Extended Reconstructed Sea Surface Temperature (ERSST) v4[37], developed by National Oceanic and Atmospheric Administration (NOAA). For data comparison, we also use other SST datasets, including NOAA ERSST v5[38], Hadley Centre SST dataset (HadSST3)[39], and Kaplan SST v2[40]. For atmospheric variables, we use National Centers for Environmental Prediction (NCEP)/National Center for Atmospheric Research (NCAR) Reanalysis 1[41]. In Fig. 2a, the climatology is calculated as a 30-year average for 1948–1977.

**Definitions of climate indices.** To quantify the impacts on the warming hole, we defined a NAWH index as average SST in the subpolar North Atlantic (45°N–65°N, 50°W–20°W) minus average SST for the broader extratropical North Atlantic (30°N–65°N, 70°W–10°W); the two regions are marked as blue boxes in Fig. 1a. This index is designed to capture regional SST deviation in the subpolar North Atlantic from the surrounding North Atlantic; such regional deviation is more relevant to the concept of the warming hole than the subpolar SST change. Relative TIO SST is defined as the tropical Indian Ocean SST (30°S–30°N, 40° E–100°E) minus mean SST for all tropical oceans (30°S–30°N).

**Coupled climate simulations.** We use a fully coupled general circulation model (GCM), the Community Earth System Model (CESM) 1.0.6[42], developed by the National Center for Atmospheric Research (NCAR). We choose the T31_g37 spatial resolution that corresponds to about 4° for the atmosphere, and 2° of latitude and 4° of longitude for the ocean with a refinement of latitudinal resolution to 0.5° in low- and high-latitude oceans. Ocean and atmosphere are fully coupled globally except for the region where we perturb ocean surface temperature.

We firstly conduct idealized simulations to demonstrate the coupled system response to the tropical Indian Ocean warming. All the simulations are integrated for 200 years unless specifically mentioned. PI refers to the control, pre-industrial simulation. In TIO + 1C, we restore the tropical Indian Ocean SST (30°S–30°N, 40° E–100°E with a 5° buffering zone at the edges inside this region; see the black box in Fig. 1) to a seasonally varying climatology that is 1 °C warmer than the control run ("PI") with a relaxation timescale of 10 days. TIO-1C and TIO + 2C are similar but with a 1 °C cooling and a 2 °C warming, respectively.

Next, we conduct two more realistic simulations using the observed tropical warming pattern (estimated as the warming trend of 1950–2015 multiplied by 66 years in ERSST v4) but retaining the model's internal variations. Specifically, for TIO + OBS, we firstly restored the tropical ocean SSTs to the warmer climatology in the tropical Indian ocean (i.e. the PI control climatology plus observed warming), averaged the restoring heat flux over the last 80 years of the 100-year integration, and then added this diagnosed seasonally varying heat flux to the Indian ocean in a new coupled simulation. The TO + OBS experiment is similar to TIO + OBS but we applied this technique to all tropical basins within 30°S–30°N (again, with a 5° buffering zone) so that all three tropical ocean basins warm by the observed amount.

More details on the experimental set-up can be found in ref. [26]. That study has investigated how Indian Ocean warming strengthens the AMOC primarily by increasing salinity in the tropical/subtropical Atlantic. Subsequently, the AMOC strength changes little in the first three decades, but then starts to increase as the salinity anomalies are transported northward. The AMOC stabilizes in about a century with a slight partial recovery in the TIO + 2C case[43,44]. The AMOC strengthening leads to the warming of subpolar North Atlantic SST, here termed as the slow response.

## Data availability

ERSST v4 is publicly available at: https://psl.noaa.gov/data/gridded/data.noaa.ersst.v4. html. ERSST v5 is publicly available at: https://psl.noaa.gov/data/gridded/data.noaa.ersst. v5.html. HadSST3 is publicly available at: https://www.metoffice.gov.uk/hadobs/hadsst3/. Kaplan SST v2 is publicly available at: https://psl.noaa.gov/data/gridded/data.kaplan_sst. html. NCEP/NCAR Reanalysis 1 is publicly available at: https://psl.noaa.gov/data/ gridded/data.ncep.reanalysis.html. The data used to plot the figures in this study are available at https://doi.org/10.5281/zenodo.3964465.

## Code availability

The model code used to impose a 1 °C Indian Ocean warming is available at https://doi. org/10.5281/zenodo.3964465.

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

## Acknowledgements

The authors thank Kaylea Nelson for her help in setting up the computational environment. Discussions with Richard Seager, Mark Cane, and Amy Clement are acknowledged. We also acknowledge computational support from the NSF/NCAR Cheyenne Supercomputing Center. This research was supported by a Lamont-Doherty Postdoctoral Fellowship to S.H., and by grants to A.V.F. from NSF (AGS-1756682, OPP-1741841) and by the Guggenheim Fellowship. Additional support is provided through the ARCHANGE project (ANR-18-MPGA-0001, France).

## Author contributions

S.H. and A.V.F. conceived and contributed equally to the study. S.H. conducted the numerical simulations, performed the data analysis, and led the writing of the manuscript. S.H. and A.V.F. together interpreted the results and edited the manuscript.

## Competing interests

The authors declare no competing interests.
