## [Peer Review File · Nature Communications]

Reviewers' comments:

Reviewer #1 (Remarks to the Author):

Review of "Indian Ocean warming as a driver of the North Atlantic warming hole" by Hu and Fedorov

This manuscript presents a suite of climate model experiments aimed at understanding the so-called North Atlantic subpolar 'warming hole' (NAWH) seen in the observed SST record over recent decades. The model experiments examine the impact of Indian Ocean or all-tropics SST anomalies on the North Atlantic SST (NASST), through atmospheric teleconnections. Various idealised or observed SST anomalies are imposed on a pre-industrial control run of the CESM climate model, and the evolution of the NASST response is analysed over 200 years.

A key result is that there are two timescales of response in the subpolar NASST: first a wind-driven 'warming hole' (cooling) response resulting from atmospheric teleconnections from the tropical SST anomalies to the subpolar Atlantic, then (emerging after a few decades) a warming response as the AMOC spins up due to tropical P-E changes. The longer term warming eventually cancels the warming hole, leading to a net warming.

The mechanisms of the teleconnection are discussed in previous publications (refs 21-23,19), while this paper focuses on demonstrating their impact on SST through the model experiments, as well as the longer term AMOC-driven response.

Previous explanations of the NAWH have largely relied on local mechanisms, and these results provide an interesting and valuable new perspective. As the authors show, just concentrating on Indian Ocean SST anomalies rather over-emphasises the mechanism shown here, but it is still present (if more muted) when representative SST anomalies are imposed across the whole tropics. In my view the authors make a good case that the observed tropical SST anomalies may have played a significant role in the variability of subpolar NASST over the past century. The results have potentially very important consequences for the use of SST patterns as a proxy for AMOC variations, which have had a lot of attention recently. They also show that the NAWH of the recent past may result from rather different mechanisms to the NAWH that appears in 21st Century climate projections.

Overall I found the paper interesting, well-written and focused and I think the material is worthy of publication in Nature Communications. I only have a few comments. Two are suggestions to improve the impact of the paper, and the other is more a comment on the authors' related paper reference 19 so may be considered out of scope here.

Comments [ref line numbers]:

[Abstract] To me, two very important consequences of the results are first, that the mechanism of the recent past NAWH may be different to the mechanism of the projected 21st Century NAWH; and secondly that the fast SST response to tropical SST anomalies shown here may act to confound SST-based proxies for multidecadal AMOC variability that have received much attention recently. These points are well discussed in the Discussion section, but reading the abstract as it stands, a busy reader might not pick up on the significance of the paper's results and choose not to read further. That would be a shame. This is for the authors' judgement, and I know the abstract is length-limited, but I would recommend that they somehow include a flag to these consequences in the Abstract.

[Paper structure: 74 onwards] After the introduction, the paper piles in directly with results. I appreciate there is value in collecting the methods together at the end so as to focus on the key

results, but the limited amount of explanation here meant that I couldn't immediately understand why the experiment was being done. The flow of the paper would be improved by one or two more sentences explaining the experiments, either at the end of the Introduction or at the start of the Results section (can still refer to Methods section for the details).

[103-105]: I wasn't convinced by this argument. The Atlantic precip response shown in Fig. 3 is an increase north of the equator and a decrease south of the equator – a bit like a northward shift of the ITCZ. The expected response to this precip anomaly would be to advect fresh anomalies northward and so *weaken* the AMOC after a few decades. This mechanism (or more typically the inverse of it in scenarios where the precip anomalies are driven by an initial AMOC weakening) has been seen in quite a few model studies over many years (e.g. Vellinga et al. 2002, J. Clim. 15, 764-780). The equator acts as a dynamical barrier so the salting south of the equator can't reach the North Atlantic so easily. I have looked at the authors' companion paper (ref 19). Ref 19 supplementary Fig 4 suggests that the positive salinity anomalies may be coming from 30N in the Atlantic rather than the tropics. The Rossby wave teleconnection is also more obvious to 30N than to the tropics (this paper, Fig.4). Anyway this is not the place to review ref. 19. I'm not doubting the results of the experiments, and whichever interpretation is right doesn't affect the results of this paper. Maybe a more neutral wording (e.g. "The Indian Ocean warming drives changes in Atlantic precipitation, and resultant salinity increases act...") would keep everyone happy?

Specific comments [line numbers]:

[51-52, 99]: This refers to a 2018 AGU talk. Not sure if this is OK to cite. Is there a paper yet?

Editorial/typos:

[Figure 1]: Please state units.

Reviewer #2 (Remarks to the Author):

General Comments:

This study explores the forcing mechanisms on the North Atlantic warming hole by performing sensitivity experiments with CESM. The authors showed that Indian ocean warming initially cools the subpolar North Atlantic by inducing a NAO-like atmospheric teleconnection in response to tropical convection. At longer time scales, AMOC strengthens which enhances northward heat transport into the subpolar North Atlantic and reverses the initial cooling in the warming hole. Thus, the warming hole is forced by a combination of atmospheric and oceanic processes at different time scales. This study is a significant contribution to understand warming hole formation mechanism by pointing out the role of the atmospheric forcing. I only have some minor comments to improve the presentation of the results.

Specific Comments:

1. This study discusses a century-long cooling trend in the North Atlantic warming hole, and the analysis of CESM experiments is separated into fast (initial 20 years) and slow (151-200 yr). However, the observed SSTA trend shown in the main text are for the 1950-2015 period. I would suggest the 1900-2015 SSTA trend (as in the Fig. S1) is shown in the main text instead. Comparing Fig. S1 and Fig. 1, I felt a cooling trend from 1900-2015 but a weak warming from 1950-2015 in the subpolar North Atlantic is consistent with the fast and slow mechanisms proposed by the authors.
2. The separation time scale (20-yr) of fast and slow response is somewhat arbitrary. According to Fig. 5c, the cold SSTA along the 40-60N persists for at least 50 years. Could the authors clarify why the

20-yr cut-off was selected.

3. Line 87-90: The conclusion is supported by Josey et al. (2019), an observational study on the causes of extremely cold wintertime SST in the Irminger Sea .

Josey, S. A., M. F. de Jong, M. Oltmanns, G. K. Moore, and R. A. Weller (2019): Extreme variability in Irminger sea winter heat loss revealed by ocean observatories initiative mooring and the ERA5 reanalysis. *Geophys. Res. Lett.* 46, 293-302.

4. Significance test should be performed for Fig. 2-4 and 6.

5. The figures can be better organized by combining Fig. 3 with Fig. 4a and moving Fig. 4b to Supplementary Materials.

Reviewer #3 (Remarks to the Author):

I have reviewed the manuscript 'Indian Ocean warming as a driver of the North Atlantic warming hole' by S. Hu and A.V. Fedorov

This is an interesting manuscript addressing the origin of the North Atlantic warming hole that appeared in SST trends of the last 70 years or so. The results presented are novel in the sense that they put forward a hypothesis for the warming hole. Namely that it was forced by a teleconnection from a strong warming trend appearing in the Indian Ocean through a fast atmospheric bridge, and not an AMOC adjustment. The study also argues that on longer time scales this teleconnection is expected to strengthen the AMOC and thus warm the North Atlantic even more. The paper is well written and rather strong on the modelling side. However, I would suggest a revision that strengthens the observational underpinning.

Major comments:

1. The conclusions of this study are almost entirely based on a modelling study. However, there are several re-analysis products around that should be good enough to look at atmospheric trends for the period 1950 to 2015. In particular, I would be interested to see the observed trends from 1950 to 2015 of the North Atlantic SLP and wind (this should be really excellent data) compared to the modelled slp and winds trend. Currently only the modeled 500 hPa height and surface wind trends are shown, but that's not enough. We need to get an idea how much of the observed trend can really be explained by the idealized model experiments. Even if it's just a fraction (which I guess it is) this would be still an interesting and novel contribution to explaining the observed trends. What is the fraction of NAO trend that can be explained by the Indian Ocean warming?

2. Since the fast response may also be due to heatfluxes, it would be very important to add a detailed analysis of the surface energy balance in the model. My guess is that most of the short term responses may be explained by changes in the latent heat fluxes, but perhaps also some fast mixed-layer processes may play a role. It is important that these mechanisms are pinpointed in details using the model output.

3. Line 57. An observed Indian Ocean trend of 1C since the 1950s is not consistent with Fig. 1, if the units of the trends are C/66 years. Is this the unit of the numbers that we see? How is the trend defined?

4. Since the Indian Ocean SSTs are so important in forcing North Atlantic responses, you could also

perform regression analysis based on re-analysis data between an Indian Ocean SST index and the North Atlantic SLP or surface winds. Does this support the findings?

Response to the reviews of the manuscript “Indian Ocean warming as a driver of the North Atlantic warming hole”

by S. Hu and A.V. Fedorov

We thank the reviewers for their comments and suggestions that led to many improvements in the paper. In particular, we have now conducted additional observational analysis and added a detailed discussion as a new subsection “Observational evidence”. The text was further edited for misprints and clarity. We feel that the revised draft is significantly improved thanks to the reviewers’ constructive comments. Our point-by-point response to the reviews is below (reviewers’ comments in italic, authors’ response in blue font).

Reviewer #1 (Remarks to the Author):

Review of “Indian Ocean warming as a driver of the North Atlantic warming hole” by Hu and Fedorov

This manuscript presents a suite of climate model experiments aimed at understanding the so-called North Atlantic subpolar ‘warming hole’ (NAWH) seen in the observed SST record over recent decades. The model experiments examine the impact of Indian Ocean or all-tropics SST anomalies on the North Atlantic SST (NASST), through atmospheric teleconnections. Various idealised or observed SST anomalies are imposed on a pre-industrial control run of the CESM climate model, and the evolution of the NASST response is analysed over 200 years.

A key result is that there are two timescales of response in the subpolar NASST: first a wind-driven ‘warming hole’ (cooling) response resulting from atmospheric teleconnections from the tropical SST anomalies to the subpolar Atlantic, then (emerging after a few decades) a warming response as the AMOC spins up due to tropical P-E changes. The longer term warming eventually cancels the warming hole, leading to a net warming.

The mechanisms of the teleconnection are discussed in previous publications (refs 21-23,19), while this paper focuses on demonstrating their impact on SST through the model experiments, as well as the longer term AMOC-driven response.

Previous explanations of the NAWH have largely relied on local mechanisms, and these results provide an interesting and valuable new perspective. As the authors show, just concentrating on Indian Ocean SST anomalies rather over-emphasises the mechanism shown here, but it is still present (if more muted) when representative SST anomalies are imposed across the whole tropics. In my view the authors make a good case that the observed tropical SST anomalies may have played a significant role in the variability of subpolar NASST over the past century. The results have potentially very important consequences for the use of SST patterns as a proxy for AMOC variations, which have

had a lot of attention recently. They also show that the NAWH of the recent past may result from rather different mechanisms to the NAWH that appears in 21st Century climate projections.

Overall I found the paper interesting, well-written and focused and I think the material is worthy of publication in Nature Communications. I only have a few comments. Two are suggestions to improve the impact of the paper, and the other is more a comment on the authors' related paper reference 19 so may be considered out of scope here.

We thank the reviewer for the encouragement and all the constructive comments. We hope the revised draft and our response below have addressed all the concerns raised by the reviewer.

Comments [ref line numbers]:

[Abstract] To me, two very important consequences of the results are first, that the mechanism of the recent past NAWH may be different to the mechanism of the projected 21st Century NAWH; and secondly that the fast SST response to tropical SST anomalies shown here may act to confound SST-based proxies for multidecadal AMOC variability that have received much attention recently. These points are well discussed in the Discussion section, but reading the abstract as it stands, a busy reader might not pick up on the significance of the paper's results and choose not to read further. That would be a shame. This is for the authors' judgement, and I know the abstract is length-limited, but I would recommend that they somehow include a flag to these consequences in the Abstract.

This is a very helpful suggestion. We agree with the reviewer and now have emphasized the two points in the abstract (see below).

“We argue that the historical NAWH can potentially be explained by such atmospheric mechanisms reliant on surface wind changes, while oceanic mechanisms related to AMOC changes become more important on longer timescales. Thus, explaining the North Atlantic temperature trends and particularly the NAWH requires considering both atmospheric and oceanic mechanisms.”

[Paper structure: 74 onwards] After the introduction, the paper piles in directly with results. I appreciate there is value in collecting the methods together at the end so as to focus on the key results, but the limited amount of explanation here meant that I couldn't immediately understand why the experiment was being done. The flow of the paper would be improved by one or two more sentences explaining the experiments, either at the end of the Introduction or at the start of the Results section (can still refer to Methods section for the details).

We have now added a subsection entitled “Experimental design” before we discuss the model results. We indeed find the flow of the paper much improved.

[103-105]: I wasn't convinced by this argument. The Atlantic precip response shown in Fig. 3 is an increase north of the equator and a decrease south of the equator – a bit like

*a northward shift of the ITCZ. The expected response to this precip anomaly would be to advect fresh anomalies northward and so *weaken* the AMOC after a few decades. This mechanism (or more typically the inverse of it in scenarios where the precip anomalies are driven by an initial AMOC weakening) has been seen in quite a few model studies over many years (e.g. Vellinga et al. 2002, J. Clim. 15, 764-780). The equator acts as a dynamical barrier so the salting south of the equator can't reach the North Atlantic so easily. I have looked at the authors' companion paper (ref 19). Ref 19 supplementary Fig 4 suggests that the positive salinity anomalies may be coming from 30N in the Atlantic rather than the tropics. The Rossby wave teleconnection is also more obvious to 30N than to the tropics (this paper, Fig.4). Anyway this is not the place to review ref. 19. I'm not doubting the results of the experiments, and whichever interpretation is right doesn't affect the results of this paper. Maybe a more neutral wording (e.g. "The Indian Ocean warming drives changes in Atlantic precipitation, and resultant salinity increases act...") would keep everyone happy?*

Thanks for the comment on our earlier, companion study, Hu and Fedorov (2019), which made us think more about the underlying mechanisms. In the precipitation change field, we indeed see a meridional dipole structure associated with the northward shift of ITCZ (Fig. 4aq). However, as we emphasized in Hu and Fedorov (2019), it is the net precipitation reduction into the tropical Atlantic, including the signals both over the ocean and the surrounding continents, that matters most for the tropical Atlantic salinity changes. In the same paper, we conducted a sensitivity test by imposing salinity anomalies over tropical South Atlantic, and we saw a significant strengthening of AMOC without imposed salinity anomalies at 40°N in the western Atlantic. It suggests that the salinity increase over the tropical South Atlantic is sufficient to explain the majority of the AMOC strengthening seen in the TIO+1C case.

We nevertheless agree with the reviewer that the salinity increase at 40°N in the western Atlantic may contribute to the AMOC strengthening too. We appreciate the reviewer's raising this point. As describing the mechanisms of the AMOC strengthening is beyond the scope of this study, we have decided to leave these details for future studies, but rephrased that sentence as,

"In short, the Indian Ocean warming suppresses the Atlantic rainfall by modifying the Walker circulation along the entire equatorial belt, and the resultant positive Atlantic salinity anomalies act to strengthen the AMOC after being advected to the subpolar North Atlantic by ocean mean circulation."

The reference, Vellinga et al. (2002), mentioned by the reviewer is insightful in explaining the recovery of AMOC strength that is also seen in our TIO+2C case. We have now briefly mentioned this partial recovery in the Methods section.

"...The AMOC stabilizes in about a century with a slight partial recovery in the TIO+2C case (Vellinga et al. 2002; Thomas and Fedorov 2019)"

Specific comments [line numbers]:

[51-52, 99]: This refers to a 2018 AGU talk. Not sure if this is OK to cite. Is there a paper yet?

We have contacted the leading author of that study and have been told that they had not submitted their paper yet. At later stages of the revision, we can either delete this sentence, which will not affect the main points of our study, or update the citation, depending on the status of that paper. We will coordinate with the editor on that.

Editorial/typos:

[Figure 1]: Please state units.

Corrected.

Reviewer #2 (Remarks to the Author):

General Comments:

This study explores the forcing mechanisms on the North Atlantic warming hole by performing sensitivity experiments with CESM. The authors showed that Indian ocean warming initially cools the subpolar North Atlantic by inducing a NAO-like atmospheric teleconnection in response to tropical convection. At longer time scales, AMOC strengthens which enhances northward heat transport into the subpolar North Atlantic and reverses the initial cooling in the warming hole. Thus, the warming hole is forced by a combination of atmospheric and oceanic processes at different time scales. This study is a significant contribution to understand warming hole formation mechanism by pointing out the role of the atmospheric forcing. I only have some minor comments to improve the presentation of the results.

We thank the reviewer for the encouragement and all the constructive comments. We hope the revised draft and our response below have addressed all the concerns raised by the reviewer.

Specific Comments:

1. This study discusses a century-long cooling trend in the North Atlantic warming hole, and the analysis of CESM experiments is separated into fast (initial 20 years) and slow (151-200 yr). However, the observed SSTA trend shown in the main text are for the 1950-2015 period. I would suggest the 1900-2015 SSTA trend (as in the Fig. S1) is shown in the main text instead.

Following the reviewer's suggestion, we have now combined the previous Fig. 1 and Fig. S1 and have shown both the 1950-2015 and 1900-2015 trends in the main text. We also added a NAWH series plot in Fig. 2c showing that the most significant reduction of the NAWH index started around 1940s, which is the main reason we focused on the period after 1950. Another reason is that most atmospheric reanalysis, including the NCEP/NCAR Reanalysis 1 used in this study, becomes available only after the 1940s.

Comparing Fig. S1 and Fig. 1, I felt a cooling trend from 1900-2015 but a weak warming from 1950-2015 in the subpolar North Atlantic is consistent with the fast and slow mechanisms proposed by the authors.

This is a very insightful point. We however note that SST data discrepancies are not small. Some datasets show a stronger cooling trend in the subpolar North Atlantic during 1900-2015 than during 1950-2015 (e.g. ERSST v4 and to a less extent ERSST v5), but some show the opposite (e.g. HadSST3 and Kaplan SST v2). So, it is uncertain whether the NAWH is more significant considering 1900-2015 or 1950-2015, even though the NAWH phenomenon itself is a robust feature across various datasets or periods. Given the data discrepancies, we decided to be more conservative by not overstating the reason for the difference between 1900-2015 versus 1950-2015 changes.

2. The separation time scale (20-yr) of fast and slow response is somewhat arbitrary. According to Fig. 5c, the cold SSTA along the 40-60N persists for at least 50 years. Could the authors clarify why the 20-yr cut-off was selected.

The reviewer is right that the cold SSTA near 40-60°N persists for about 40 years and then starts to vanish (Fig. 7b). We have modified our average to the first 40 years for the fast response. We have now also clarified the choice of 40 years at L131-135.

“Here we regard the average of Years 1-40 as reflecting the fast ocean response due to the induced atmospheric changes; the 40-year duration is chosen as a compromise to be sufficiently long for statistically significant results but short enough for the generated NAWH not to decrease yet in response to AMOC slowdown. The main conclusions presented below do not change if a shorter period (e.g. 20 or 30 years) is chosen for averaging.”

3. Line 87-90: *The conclusion is supported by Josey et al. (2019), an observational study on the causes of extremely cold wintertime SST in the Irminger Sea .*

Josey, S. A., M. F. de Jong, M. Oltmanns, G. K. Moore, and R. A. Weller (2019): Extreme variability in Irminger sea winter heat loss revealed by ocean observatories initiative mooring and the ERA5 reanalysis. Geophys. Res. Lett. 46, 293-302.

Thanks for referring us to this observational study. We have now cited in the text at L59-60.

4. *Significance test should be performed for Fig. 2-4 and 6.*

In response to the reviewer’s suggestion, we have conducted significance tests to those results. The results with p-values above 0.05 are overlaid with grey hatches.

5. *The figures can be better organized by combining Fig. 3 with Fig. 4a and moving Fig. 4b to Supplementary Materials.*

Thanks for the suggestion. We have combined the SST and geopotential height (zonal-mean removed) plots as a new figure and moved the original geopotential height (zonal-mean retained) plot to the supplement. For the geopotential height field, we now use ~200 mb for better illustration.

Reviewer #3 (Remarks to the Author):

I have reviewed the manuscript 'Indian Ocean warming as a driver of the North Atlantic warming hole' by S. Hu and A.V. Fedorov

This is an interesting manuscript addressing the origin of the North Atlantic warming hole that appeared in SST trends of the last 70 years or so. The results presented are novel in the sense that they put forward a hypothesis for the warming hole. Namely that it was forced by a teleconnection from a strong warming trend appearing in the Indian Ocean through a fast atmospheric bridge, and not an AMOC adjustment. The study also argues that on longer time scales this teleconnection is expected to strengthen the AMOC and thus warm the North Atlantic even more. The paper is well written and rather strong on the modelling side. However, I would suggest a revision that strengthens the observational underpinning.

We really appreciate the reviewer's encouragement and all the constructive comments. Thank you. We hope the revised draft and our response below have addressed all the concerns raised by the reviewer.

Major comments:

1. The conclusions of this study are almost entirely based on a modelling study. However, there are several re-analysis products around that should be good enough to look at atmospheric trends for the period 1950 to 2015. In particular, I would be interested to see the observed trends from 1950 to 2015 of the North Atlantic SLP and wind (this should be really excellent data) compared to the modelled slp and winds trend. Currently only the modeled 500 hPa height and surface wind trends are shown, but that's not enough. We need to get an idea how much of the observed trend can really be explained by the idealized model experiments. Even if it's just a fraction (which I guess it is) this would be still an interesting and novel contribution to explaining the observed trends. What is the fraction of NAO trend that can be explained by the Indian Ocean warming?

This is a very helpful suggestion. The previous version of our manuscript included only a limited amount of observational analysis as our motivation. Following the reviewer's suggestions, we have now added more comprehensive observational analysis, for example on sea level pressure and surface wind patterns and variability (Fig. 2a,b), the temporal relationship between the NAWH and the overlying surface zonal winds (Fig. 2c), surface energy budget (Fig. 3; Supplementary Fig. 1), etc. This analysis has led to several new and interesting results.

Fig. 2c shows a very tight connection between the NAWH index and the North Atlantic surface zonal winds ($r = -0.90$ for timescales longer than decadal), which provides direct observational evidence that the historically observed NAWH may have been driven by atmospheric processes. Fig. 3 shows that even though the surface wind strengthening over the subpolar North Atlantic (Fig. 2a,b) tends to enhance surface turbulent heat fluxes, such effect is difficult to detect in the actual surface turbulent heat flux changes that are dominated by the thermal damping effect (i.e. the cooling of SST damps surface turbulent heat fluxes). It suggests that an analysis on the surface heat flux change alone that mixes

the forcing and feedbacks might sometimes provide limited or even misleading information on the origin of surface temperature changes.

We have now added a comprehensive discussion, entitled “Observational evidence”, for those results, which we think have greatly improved our manuscript.

Also, we tried to estimate the fraction of the observed changes in the NAWH and the subpolar surface zonal winds (an index for NAO but more relevant to the NAWH) that could be explained by the Indian Ocean warming. The results are presented in Fig. 9a and Fig. 5c, respectively. In short, we find that, based on our model estimates, the observed relative TIO warming since 1950 could explain ~87% of the NAWH and ~55% of the subpolar surface zonal wind changes. We stress however that those are only rough estimates given the observational data uncertainty and model dependence, for example. We have discussed this part of results in the Discussion section (L263-267).

2. Since the fast response may also be due to heatfluxes, it would be very important to add a detailed analysis of the surface energy balance in the model. My guess is that most of the short term responses may be explained by changes in the latent heat fluxes, but perhaps also some fast mixed-layer processes may play a role. It is important that these mechanisms are pinpointed in details using the model output.

In terms of the processes that drive the subpolar cooling, we think both the enhanced surface turbulent heat fluxes due to the wind strengthening and the southward Ekman transport of climatologically cold high-latitude water due to the westerly anomalies could play a role. This is consistent with, for example, Marshall et al. (2001). We have now clarified the two processes in both the observational part (L74-76) and the modeling part (L150-152; L156-157).

In response to the reviewer’s comments, we have analyzed surface heat flux changes over the NAWH and found that the net change of surface turbulent heat fluxes is downward, heating the ocean (Supplementary Fig. 3). This is however not inconsistent with the argument the reviewer mentioned, also as we put forward in the paper, that surface wind strengthening would enhance surface turbulent heat fluxes and thus cool the mixed-layer ocean. As we discussed in the observational analysis, the actual surface turbulent heat flux changes are dominated by the thermal damping effect due to SST decrease (and thus reduced turbulent heat flux, which is a warming effect for the ocean). Therefore, the attribution based on surface energy budget analysis could sometimes be obscured by the mixture of all forcings and feedbacks.

We have now added a comprehensive discussion on the surface energy budget and potential caveats in the main text. We have done so mainly for the observational part, which we think is more compelling (L78-90; Fig. 3; Supplementary Fig. 1), and also we have re-emphasized this point briefly for the modeling results (L152-158; Supplementary Fig. 3).

3. Line 57. An observed Indian Ocean trend of 1C since the 1950s is not consistent with Fig. 1, if the units of the trends are C/66 years. Is this the unit of the numbers that we see? How is the trend defined?

We apologize for this confusion caused by missing units in Fig. 1, which should be °C/decade. We have now added the units in the figure caption. The long-term trends are calculated based on a linear regression. The Indian Ocean SST trend is about 0.15°C/decade on average, which corresponds to ~1°C within 66 years.

4. Since the Indian Ocean SSTs are so important in forcing North Atlantic responses, you could also perform regression analysis based on re-analysis data between an Indian Ocean SST index and the North Atlantic SLP or surface winds. Does this support the findings?

Thanks for the suggestion. We have now computed a linear regression of North Atlantic sea level pressure and surface winds onto the relative TIO SST, and the results are presented in Supplementary Fig. 4. It indeed shows an intensified surface zonal winds at the south of Greenland, where the NAWH is located, associated with a positive relative TIO SST. We have now discussed this analysis at L255-258.

REVIEWER COMMENTS

Reviewer #1 (Remarks to the Author):

Thank you to the authors for their revision of the manuscript including consideration of my previous comments. All my concerns and suggestions have been well addressed in the revised version. I also think the inclusion of reanalysis results in response to another reviewer has further strengthened the paper.

I think this is an important and valuable piece of work and am very pleased to recommend publication.
Richard Wood

Reviewer #2 (Remarks to the Author):

The authors have satisfactorily addressed my comments. I found the manuscript improved, especially the discussion of heat flux trend, and thus recommend its acceptance.

Reviewer #3 (Remarks to the Author):

I am overall mainly satisfied with the revisions. However, since the main mechanism proposed for the short-term North Atlantic cooling is now Ekman transport, I encourage for the sake of the reader to perform an estimate in terms of C/day of this effect and to compare it with the Q_{net} term. At least for the model such an estimate should be possible based on surface winds and upper ocean temperature structure. I therefore recommend another minor revision.

**Response to the 2nd round of reviews of the manuscript
“Indian Ocean warming as a driver of the North Atlantic warming hole”**

by S. Hu and A.V. Fedorov

We thank the reviewer for the further suggestion. We have now added a discussion on the rough estimate of the anomalous Ekman transport-induced heat flux using reanalysis datasets. We feel that the revised manuscript is further improved and provides additional insights with more quantitative arguments. Our point-by-point response to the reviewer’s comments is below (reviewer’s comments in *italic*, authors’ response in blue font).

Reviewer #3 (Remarks to the Author):

I am overall mainly satisfied with the revisions. However, since the main mechanism proposed for the short-term North Atlantic cooling is now Ekman transport, I encourage for the sake of the reader to perform an estimate in terms of C/day of this effect and to compare it with the Q_{net} term. At least for the model such an estimate should be possible based on surface winds and upper ocean temperature structure. I therefore recommend another minor revision.

Following the reviewer’s suggestion, we have now conducted a quantitative estimate of the ocean heat flux due to the anomalous Ekman transport. We obtained this estimate by using the equation $Q'_{Ek} = (c_p \tau'_x / f) (\partial T / \partial y)$, where c_p is the specific heat of seawater, τ'_x is the surface zonal wind stress change, f is the Coriolis parameter, and $\partial T / \partial y$ is the climatological meridional SST gradient. Here we assume $c_p = 3850 \text{ J kg}^{-1} \text{ }^\circ\text{C}^{-1}$, $f = 1.1 \times 10^{-4} \text{ s}^{-1}$, and $\partial T / \partial y = -4 \times 10^{-3} \text{ }^\circ\text{C km}^{-1}$ for the region of interest. The long-term change of τ'_x is about $+0.02 \text{ N/m}^2$ since 1950, based on NCEP NCAR Reanalysis 1, and it would give rise to a negative Q'_{Ek} of about -3 W/m^2 , the magnitude of which is comparable to or greater than the positive net ocean surface heat flux change Q'_{net} over the subpolar North Atlantic (Fig. 3b). Here it is just a rough estimate and a more accurate attribution requires a closer look into the oceanic processes (probably using ocean assimilation datasets) and a more comprehensive calculation of ocean heat budget. Also note that, besides Q'_{Ek} , other oceanic processes (e.g. buoyancy-driven ocean circulation change, passive advection of the mixed layer ocean temperature change, etc.) could also give rise to ocean heat flux changes and therefore balance Q'_{net} . If we assume a mixed layer ocean depth of 100 m, a Q'_{Ek} of -3 W/m^2 is equivalent to a cooling effect of about -0.2°C/year . This is one order of magnitude greater than the temperature tendency since 1950, implying that the mixed layer ocean temperature is in quasi-equilibrium with the atmosphere.

We have now added the quantitative estimate and the relevant discussion at L73-75 and L83-89.

REVIEWERS' COMMENTS:

Reviewer #3 (Remarks to the Author):

The authors have addressed the remaining issue satisfactorily.
Therefore I recommend to accept this manuscript.